# ROYAL SOCIETY
# OPEN SCIENCE

mathematical modelling/statistics/integral equations

particle method, Bayesian inverse problems, non-local cancer model, proliferation function, stability of posterior distribution, parameter estimation

**Author for correspondence:**
Zuzanna Szymańska
e-mail: z.szymanska@icm.edu.pl

# Bayesian inference of a non-local proliferation model

Zuzanna Szymańska[1,3], Jakub Skrzeczkowski[2],
Błażej Miasojedow[2] and Piotr Gwiazda[3]

[1]ICM, University of Warsaw, ul. Tyniecka 15/17, 02-630 Warsaw, Poland
[2]Faculty of Mathematics, Informatics and Mechanics, University of Warsaw, ul. Banacha 2, 02-097 Warsaw, Poland
[3]Institute of Mathematics, Polish Academy of Sciences, ul. Śniadeckich 8, 00-656 Warsaw, Poland

 ZS, 0000-0002-2968-5627

From a systems biology perspective, the majority of cancer models, although interesting and providing a qualitative explanation of some problems, have a major disadvantage in that they usually miss a genuine connection with experimental data. Having this in mind, in this paper, we aim at contributing to the improvement of many cancer models which contain a proliferation term. To this end, we propose a new non-local model of cell proliferation. We select data that are suitable to perform Bayesian inference for unknown parameters and we provide a discussion on the range of applicability of the model. Furthermore, we provide proof of the stability of posterior distributions in total variation norm which exploits the theory of spaces of measures equipped with the weighted flat norm. In a companion paper, we provide detailed proof of the well-posedness of the problem and we investigate the convergence of the escalator boxcar train (EBT) algorithm applied to solve the equation.

## 1. Introduction

Mathematical models of complex biological phenomena that are developed nowadays are based on the knowledge of biophysical processes. Thanks to this, we correctly obtain the general structure of equations, but unfortunately, we miss the information on parameter values. This is obviously mainly due to the difficulty of performing specific experimental measurements. It is important to underline that sometimes there are no established experimental scenarios with which one can measure the desired parameter values, to say nothing about the errors related to measurements when possible. The lack of reliable values of the parameters implies a significant limitation of the applicability of those models. To overcome this difficulty, the approach taken so far was to search for the model parameters in the literature. However, these were usually deficient or obtained within

particular experimental regimes, very often not corresponding to the scenario under consideration. Therefore, it is not clear at all that their interpretation is like the parameters in the mathematical model.

Many, if not most, cancer models contain some type of logistic function to describe cancer cell proliferation (see [1–5] and references therein). Mainly due to its conceptual ease, this approach appears to researchers as a tempting one. Unfortunately, this description has important drawbacks. First of all, to capture the spatial expansion of the colony, terms like diffusion or different types of taxis are added, even if they are not biologically justified. Moreover, those models are usually short of reliable values of the parameters. Within this paper, we aim at improving potential cancer models, in particular, cancer invasion models, by proposing a new description of proliferation. More precisely, we extend the classical logistic proliferation function to include a non-local integral term in the growth part. Such modification allows capturing the spatial expansion, i.e. appearance of cells in new locations, without adding artificial terms. To assess the usability of the new model, we apply the inverse problem methodology, i.e. we provide a Bayesian inference for unknown parameters and we demonstrate the accuracy of estimators on experimental data on multicellular spheroids growths for three different cells lines. We determine the reliable range of applicability of the newly proposed model. Since the Bayesian approach requires a large number of simulations, we use the fact that, under some assumptions, the proposed model is radially symmetrical, and we transform it into spherical coordinates. Finally, we prove the stability of the numerical scheme used to solve the model and we give conclusions.

The structure of the paper is as follows. In §2, we introduce the new mathematical model, a non-local proliferation function of the logistic type, and compare it to a previously proposed one. Then, we present the experimental data that we found most suitable for parameter estimation of simple cellular colony growth. We conclude §2 with theoretical consideration on the range of the model applicability. In §3, we briefly describe the adopted Bayesian inference methodology and we present the observation model. In §4, we prove stability of posterior distributions. Then in §5, we show the results of parameter estimation, and finally, in §6, we discuss the obtained results and direction of future research. For the interested reader, we include appendix A and appendix B, one containing an auxiliary lemma on the Lipschitz continuity of the inverse function, and the second one containing results on the convergence of measure solutions of the considered problem (we also refer to our companion paper for more analytical details [6]). Appendix C contains the pseudo-code for the random walk Metropolis–Hastings algorithm.

# 2. Non-local proliferation model and experimental data

A proper description of cell colony development is a very difficult task. This is mainly because many factors influence those dynamics. Even when considering the increase in cell number alone, in addition to cell division an important role is played by access to nutrients and population density. The latter particularly affects the dynamics of larger cell colonies, being significantly less important in the development of colonies at the initial stage. Moreover, the overlap of so many phenomena causes the setting up of an experimental regime suitable for obtaining data for parameter estimation of proliferation function to be very challenging.

An approach that researchers use frequently to picture the increase in cell number is the logistic function, which is sometimes modified, for example, by adding volume filling term. However, as already mentioned, the simple local logistic function applied to describe the proliferation of cells within a colony is unable to capture its spatial expansion. This problem is often bypassed by the inclusion of artificial terms, primarily diffusion, and different types of taxis, making the description more phenomenological and more distant from the biological process.

Therefore, we propose a new non-local logistic function to describe the proliferation of cells living within a colony where the integral term is introduced in the growth part to capture the phenomena of the emergence of daughter cells adjacent to proliferating cells, i.e.

$$\partial_t n(x, t) = \alpha k * n(x, t) \left(1 - n(x, t)\right), \tag{2.1}$$

where $\alpha$ stands for proliferation rate, and $k = k(x)$ is a kernel function with compact support such that,

$$k * n(x, t) = \int_{\mathbb{R}^3} k(x - y) n(y, t) \, \mathrm{d}y.$$

We assume that $k(x)$ is fixed in time radially symmetric kernel with profile $K$ that is $k(x) = K(|x|)$ for $x \in \mathbb{R}^3$. An interesting issue is the choice of a particular shape of kernel $k$. In our approach, we choose a kernel corresponding to a normalized characteristic function of a ball, i.e.

$$K(|x|) = \frac{3}{4\pi} \sigma_k^{-3} 1_{[0,\sigma_k]}(|x|), \qquad (2.2)$$

where $\sigma_k$ stands for kernel size. Note that the inhibitory term $(1 - n(x, t))$ remains local according to the interpretation that the emergence of a cell in a given location is limited by the density of cells in that very place. To the best of our knowledge, a non-local logistic proliferation function given by (2.1) was not proposed before. However, another type of non-local logistic proliferation function was published previously by Maruvka & Shnerb in 2006, who suggested including an integral term in the inhibitory part [7]. Importantly, in their approach, the colony progression in space is again induced solely by the diffusion term that they keep in their model. Finally, we would like to mention that in previous literature other interesting approaches described the dynamics of multicellular spheroids, for instance, proposed by Byrne and Chaplain [8–10] and later analysed in many analytical papers [11–14].

It is intuitively clear that the proposed model (2.1) describes the dynamics of the cellular colony whose local maximal density is limited by the carrying capacity of the environment, and spatial progression is driven by the emergence of new cells in the neighbourhood of dividing mother cells. However, to reliably assess the model and possibly determine its range of applicability we need to refer to real data. Undoubtedly, the best data are those obtained within an experimental regime limiting the influence of phenomena other than proliferation itself to the biggest possible extent.

Despite being quite old, from that perspective probably the best data are the classical data published by Folkman & Hochberg in 1973 showing the evolution of multicellular spheroids in laboratory conditions with the medium being replenished and open space made available [15]. The experiments were carried out for three different cell lines, i.e L-5178Y murine leukaemia cells, V-79 Chinese hamster lung, and B-16 mouse melanoma. Even though the experiments were carried out for three different cell lines the cultivated spheroids experienced the same general growth pattern. Figure 1 shows the dynamics of mean diameter and standard deviation of cultivated spheroids of L-5178Y, V-79 and B-16 cell lines, redrawn from the original paper. At first, the spheroids enlarged exponentially for a few days before the onset of central necrosis and then, for several weeks, continued on linear growth beginning with the appearance of necrosis [15]. After reaching a critical diameter the spheroids experience no further expansion [15]. Although the general growth pattern was the same, the precise age of switching between exponential and linear growths and stabilization differed for cell lines under investigation.

We now turn to a short theoretical discussion on the possible range of applicability of the model (2.1). According to Folkman & Hochberg, the exponential growth of spheroids lasts for several initial days only [15]. Then increase in colony volume is (approximately) proportional to its volume, and this is because initially all cells divide regardless of where they are located in the spheroid. However, this period in colony development continues as long as the cell number is relatively small and therefore might be beyond the scope of applicability of a density model—for that initial period of colony growth, the discrete description is probably more adequate. Since the appearance of the central necrosis, the proliferation within the spheroid is limited to the outer layer of several rows of cells. That means that the colony growth becomes (again approximately) proportional to its surface area. Assuming the range of the kernel $k(x)$ to be significantly smaller than the radius of the spheroid, then such a scenario is suitable to describe with (2.1). Finally, to explain the observed cease of growth of spheroids we theorize that, over time, their dynamics become influenced by processes that are not present at the beginning of cultivation, which go beyond the phenomenon of proliferation. This may be, for example, the lysis of the necrotic part of the spheroid or an inhibitory effect resulting from the appearance of catabolites or necrotic debris.

Taking the above into account, we formulate a hypothesis that the model given by (2.1) is suitable to describe the evolution of cell colonies in which the growth is limited to the outer layer of cells. To bolster the hypothesis, we confront the model with experimental data presented on figure 1 and, for the period of quasi-linear growth of the tumour diameter, we perform parameter estimation using the Bayesian inference approach. Although it seems a natural choice, the kernel given by (2.2) causes some analytical and practical problems. To solve (2.1) equipped with compactly supported and Lipschitz continuous kernel one could apply an approach based on the numerical scheme called escalator boxcar train (EBT) developed by de Roos [16]. However, the low regularity of the kernel which is not Lipschitz continuous prevents using standard arguments to prove the convergence of the algorithm.

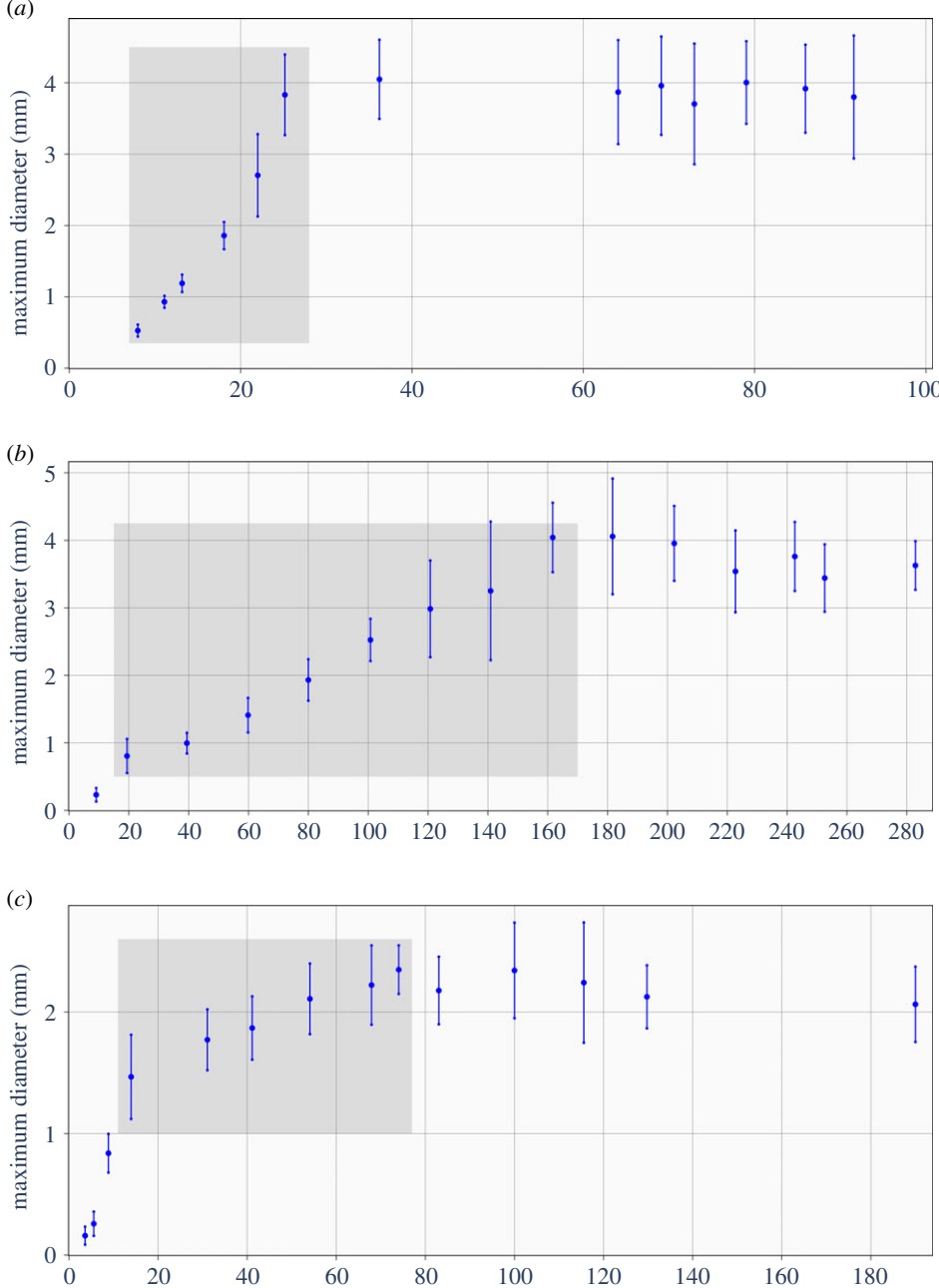

**Figure 1.** Evolution of multicellular spheroids diameters of three cell lines grown *in vitro* in unlimited fresh medium and space. Spheroidal growth appears in three stages: (i) a brief phase of exponential growth before the onset of central necrosis; (ii) a linear growth beginning with the appearance of necrotic cells in the centre of spheroids; and (iii) a dormant phase that begins when spheroid ceases to expand further [15]. Grey rectangles indicate the time span of linear expansion. Except for the grey rectangles, data are rewritten from Folkman & Hochberg [15]. (*a*) Mean diameter and standard deviation of 50 isolated spheroids of L-5178Y cells. (*b*) Mean diameter and standard deviation of 70 isolated spheroids of V-79 cells. (*c*) Mean diameter and standard deviation of 32 isolated spheroids of B-16 cells.

Therefore, we use the fact that both initial data and kernel $k(x)$ are spherically symmetrical, and we rewrite (2.1) using spherical coordinates that gives us (2.3). We refer the reader interested in the detailed arithmetic of the change of variables to the appropriate section in our companion paper [6].

Let $n(x, t)$ be the solution to (2.1) with radially symmetric initial condition $n_0(x)$. Then the radial density $p(R, t)$ defined as

$$p(R, t) = 4\pi R^2 \, n((0, 0, R), t)$$

with $p_0(R) = 4\pi R^2\, n_0((0,\, 0,\, R))$ satisfies

$$\partial_t p(R,\, t) = (4\pi R^2 - p(R,\, t)) \int_0^\infty L(R,\, r) p(r,\, t)\, \mathrm{d}r, \tag{2.3}$$

where the interaction kernel $L(R,\, r)$ is given by

$$L(R,\, r) = \frac{3\alpha}{16\,\pi\,\sigma_k^3} \frac{\min\{(R+r)^2,\, \sigma_k^2\} - \min\{(R-r)^2,\, \sigma_k^2\}}{R\,r}. \tag{2.4}$$

Equation (2.3) has Lipschitz kernel with only one singularity at zero. Using appropriate weighted norms, we have proven that the numerical algorithm based on EBT approach converges in the setting of Radon measures. Since the full proof of this fact goes beyond the scope of this paper again we refer to appendix B and our analytical paper for more details [6]. Here, let us only remark that the EBT, or more general particle methods, for (2.3) boils down to the following. As the solutions to (2.3) are supported for all $R \geq 0$ even if one starts with compactly supported initial conditions, we introduce $R_0 > 0$ such that $p(R, t)$ is negligibly small for $R \geq R_0$ (see theorems B.1 and B.5 for the precise statement), and we approximate the distribution

$$p(R,\, t) \approx \sum_{i=1}^{N} m_i(t)\, \delta_{x_i}, \tag{2.5}$$

where $x_i = \frac{i}{N} R_0$ and $i = 1, \ldots, N$. With these assumptions in place, it is now sufficient to solve the system of ODEs for masses $m_i(t)$:

$$\partial_t m_i(t) = \left(4\pi x_i^2 \frac{R_0}{N} - p(x_i,\, t)\right) \sum_{j=1}^{N} L(x_i,\, x_j) m_j(t), \tag{2.6}$$

where $m_i(0)$ are chosen so that $\sum_{i=1}^{N} m_i(0)\, \delta_{x_i}$ approximates the initial distribution, i.e.

$$m_i(0) = \int_{x_{i-1}}^{x_i} p(r,\, 0)\, \mathrm{d}r. \tag{2.7}$$

Techniques used to prove the convergence of EBT-based numerical scheme for (2.3) involve the notion of the flat norm on the spaces of measures, which provides explicit convergence error of approximation, cf. theorem B.5, and therefore is suitable for studying the order of convergence. This has direct application to our problem as posterior distributions in our work are computed based on the numerical solutions rather than explicit ones. In general, this may result in errors but, thanks to estimates on errors of numerical approximation, we are able to prove the stability of posterior distributions. An additional benefit of the change of variables is the improvement in computational accuracy as well as the speed-up of the numerical simulations, which is particularly important considering that Bayesian inference usually requires thousands of iterations of solutions to the estimated model. We conclude the section with the remark that techniques based on the flat norms on spaces of measures became recently a promising tool for optimal control problems [17–19] which may result in the future in combining Bayesian techniques with optimal control.

# 3. Observation model

To estimate parameters of the model (2.1), we propose to use Bayesian inference. Within this approach, unknown parameters are treated as random variables that can be described with probability distributions. Bayesian inference is based on posterior distribution and the conditional distribution of parameters given the observed data. The posterior distribution, by Bayes theorem, is given by

$$\pi(\theta|D) = \frac{\pi(\theta)\ell(D|\theta)}{\int_\Theta \pi(\theta)\ell(D|\theta)\, \mathrm{d}\theta}, \tag{3.1}$$

where $D$ denotes collected data and $\theta$ denotes a given vector of parameters, whereas $\Theta$ is the space of all parameters. To be precise, $\pi(\theta\,|\,D)$ is the posterior probability density that is the probability density of $\theta$ given data $D$, $\pi(\theta)$ is the prior probability density, that is the probability density of $\theta$ without any knowledge on data, finally, $\ell(D\,|\,\theta)$ is the likelihood function that quantifies the probability of observing data $D$, given the parameter $\theta$.

Usually, it is not possible to obtain an analytical formula of the joint posterior distribution $\pi(\theta \mid D)$ given by (3.1). A possible way to overcome this difficulty is to use numerical methods, of which the very popular are the Markov chain Monte Carlo (MCMC) methods [20,21]. MCMC methods comprise a whole class of algorithms including one of the most widely used—the Metropolis–Hastings algorithm. The Metropolis–Hastings algorithm can be used to generate a sample from the posterior distribution $\pi(\theta \mid D)$, which in turn can be used to determine estimators of parameters. The main idea behind that algorithm is to simulate a Markov chain whose stationary distribution is $\pi(\theta \mid D)$. This means that for a sufficiently large number of steps, samples from the Markov chain look like the samples form $\pi(\theta \mid D)$.

The first state of the Markov chain, $\theta_0$, is selected according to some chosen *a priori* distribution $\pi(\theta)$. *A priori* distributions reflect our belief about the nature of the estimated parameters. Such a belief may be based on intuition, experience, assumptions, or even a simple guess. Then, the next step in the Metropolis–Hastings algorithm is selecting a candidate for the next state of the Markov chain taking into account the current state $\theta_j$. This requires defining a method of sampling the parameter space $\Theta$, i.e. requires defining a probability density $g(\tilde{\theta} \mid \theta_j)$, sometimes referred to as the proposal density or jumping distribution, that suggests a candidate $\tilde{\theta}$ for the next sample value $\theta_{j+1}$, given the previous sample value $\theta_j$. In the case of an unknown parameter being a number, the probability density $g(\tilde{\theta} \mid \theta_j)$ is often chosen to be a normal distribution. When unknown parameters constitute a vector, the probability density $g(\tilde{\theta} \mid \theta_j)$ is usually a multivariate normal distribution, which makes the proposing of a candidate for a new state from a current one very simple. Having the candidate for the next state of the Markov chain we have to decide whether to accept it or not. There is no single criterion for doing that; however, widely used is the function proposed by Metropolis, i.e.

$$A = \min\left\{ 1, \frac{\pi(\tilde{\theta} \mid D) g(\theta_j \mid \tilde{\theta})}{\pi(\theta_j \mid D) g(\tilde{\theta} \mid \theta_j)} \right\}. \tag{3.2}$$

In the case when $g(\theta_j \mid \tilde{\theta}) = g(\tilde{\theta} \mid \theta_j)$, i.e. the proposal density is symmetrical, (3.2) simplifies to

$$A = \min\left\{ 1, \frac{\pi(\tilde{\theta} \mid D)}{\pi(\theta_j \mid D)} \right\}. \tag{3.3}$$

Using (3.1), we obtain

$$A = \min\left\{ 1, \frac{\pi(\tilde{\theta}) \ell(D \mid \tilde{\theta})}{\pi(\theta_j) \ell(D \mid \theta_j)} \right\}. \tag{3.4}$$

Function $A$, often called the acceptance probability, gives the probability of the candidate $\tilde{\theta}$ being accepted as the next state in Markov chain. The Metropolis–Hastings algorithm generates a uniform random number $u \in [0, 1]$ and if $u \le A$ then sets $\theta_{j+1} := \tilde{\theta}$, otherwise sets $\theta_{j+1} := \theta_j$.

To calibrate the proposed proliferation model, we need to set up the link between the data and the theoretical framework, namely, we need to define the so-called observation model. To estimate the parameters, we use three series of measurements of diameters of multicellular spheroids provided by Folkman & Hochberg [15]. Assuming that all spheroids have almost homogeneous mass, we presume that colony radius at time $t$ determines the sphere containing 95% of the current mass of the spheroid, i.e.

$$r(t) = \inf\left\{ s : \int_0^s p(r, t)\, dr > 0.95 \cdot \int_0^\infty p(r, t)\, dr \right\}, \tag{3.5}$$

where $r(t)$ denotes the colony radius at time $t$, whereas $\int_0^\infty p(r, t)\, dr$ stands for colony mass at time $t$. Continuity of such a defined quantile function strongly depends on measure $p$. On the subset of measures having density with connected support, it can be shown that the quantile is Lipschitz continuous with respect to the underlying measure. This property is crucial for showing the stability of posterior distribution. In our approach, the solution is given by a discrete measure $\mu_t^N$, so to make the solution continuous, one may convolve it for instance with Laplace distribution

$$\rho_\epsilon = \frac{1}{2\epsilon}\, e^{-(|x|/\epsilon)}. \tag{3.6}$$

The comparison computations we conducted indicate that such a regularization is not needed in practice. The differences between the simulation results are imperceptible and concern distant decimal places. On

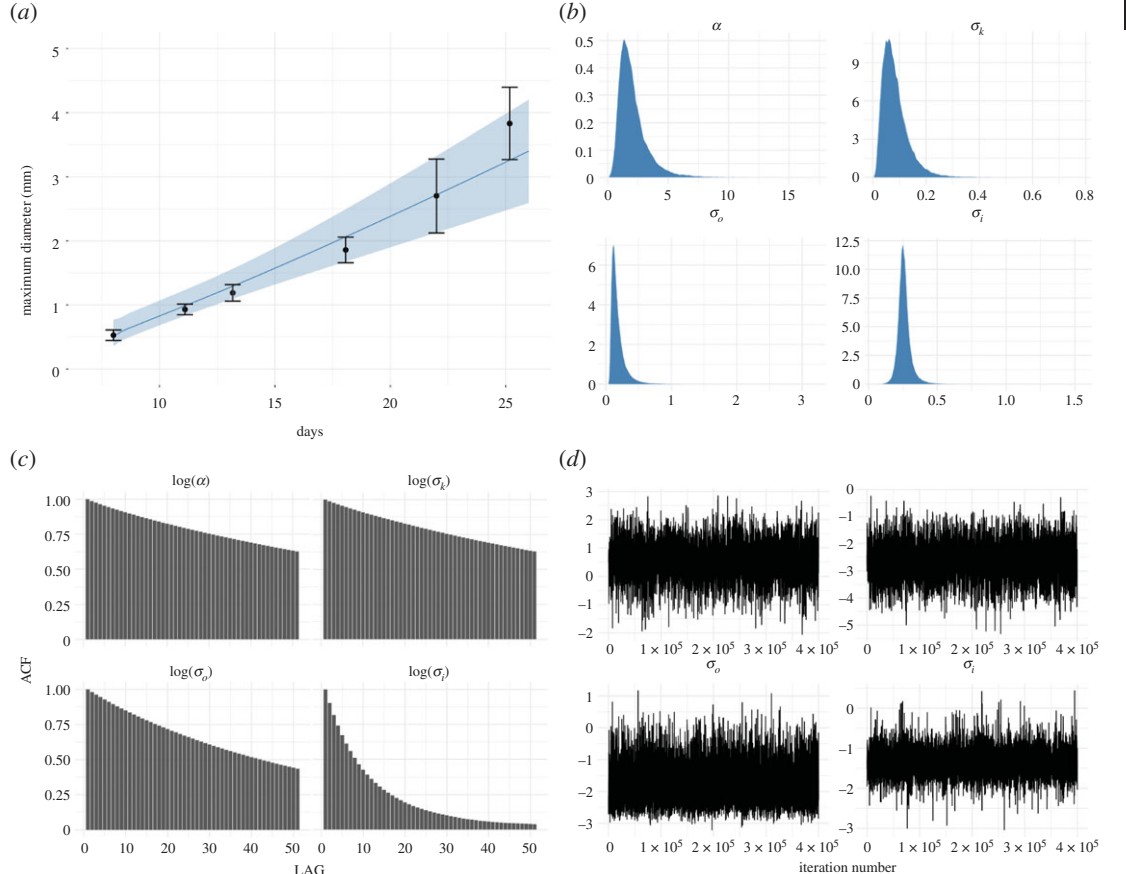

**Figure 2.** The figure shows the results of Bayesian inference for model (2.1) using the data on L-5178Y cells provided by Folkman & Hochberg ([15], or figure 2a). Plots with (a) label show the predicted mean diameters of spheroids, with black dots standing for measurements, the blue line presenting the diameters predicted by the model, and finally, the shadow area indicating the 95% credibility intervals for the predictions. Plots with (b) labels present the marginal posterior distributions of estimated parameters, whereas plots with labels (c) show the auto-correlation and plots with labels (d) stand for the trace plots of the trajectories of a random walk Metropolis. Simulations performed for $\tilde{\sigma}_i = 1.065 \cdot \sigma_i$ and $q = 13$, cf. (3.9).

the other hand, the regularization requires much more computing power, in particular, it extends the calculation many times, therefore we omit it while performing the parameter estimation.

Despite any efforts, the measurements of the spheroids diameters are of course burdened with an error, therefore we assume that

$$r_o^i = r(t_i) \cdot Z_i, \quad \text{with } \log(Z_i) \sim N(0, \sigma_o^2), \tag{3.7}$$

or alternatively

$$\log(r_o^i) = \log(r(t_i)) + \tilde{Z}_i, \quad \text{with } \tilde{Z}_i \sim N(0, \sigma_o^2), \tag{3.8}$$

where $r_o^i$ stands for colony radius at measurement performed at time $t_i$, $r(t_i)$ is the actual colony radius at time $t_i$ and $\sigma_o$ stands for homogeneous over time measurement error. In conclusion, we perform parameter estimation for three data series $D = \{r_o^i\}_{i=1}^l$, where $l$ stands for number of measurements taken into consideration.

The choice of the function describing the initial distribution is to some extent arbitrary. It seems reasonable to assume that the initial function is close to the characteristic function of the ball with the phenomenological modification consisting of mollifying the edge to capture the fact that the cell density on the colony surface is smaller than inside. We assume that it is given by

$$p(r, 0) = 4\pi r^2 \left(1 - \left(\frac{r}{\tilde{\sigma}_i}\right)^q\right) 1_{[0,\tilde{\sigma}_i]}(r), \tag{3.9}$$

where $\tilde{\sigma}_i$ and $q$ are chosen so that the radii of the initial colony calculated according to the formula (3.5) are close to $\sigma_i$, see captions to figures 2, 3 and 4 for precise values.

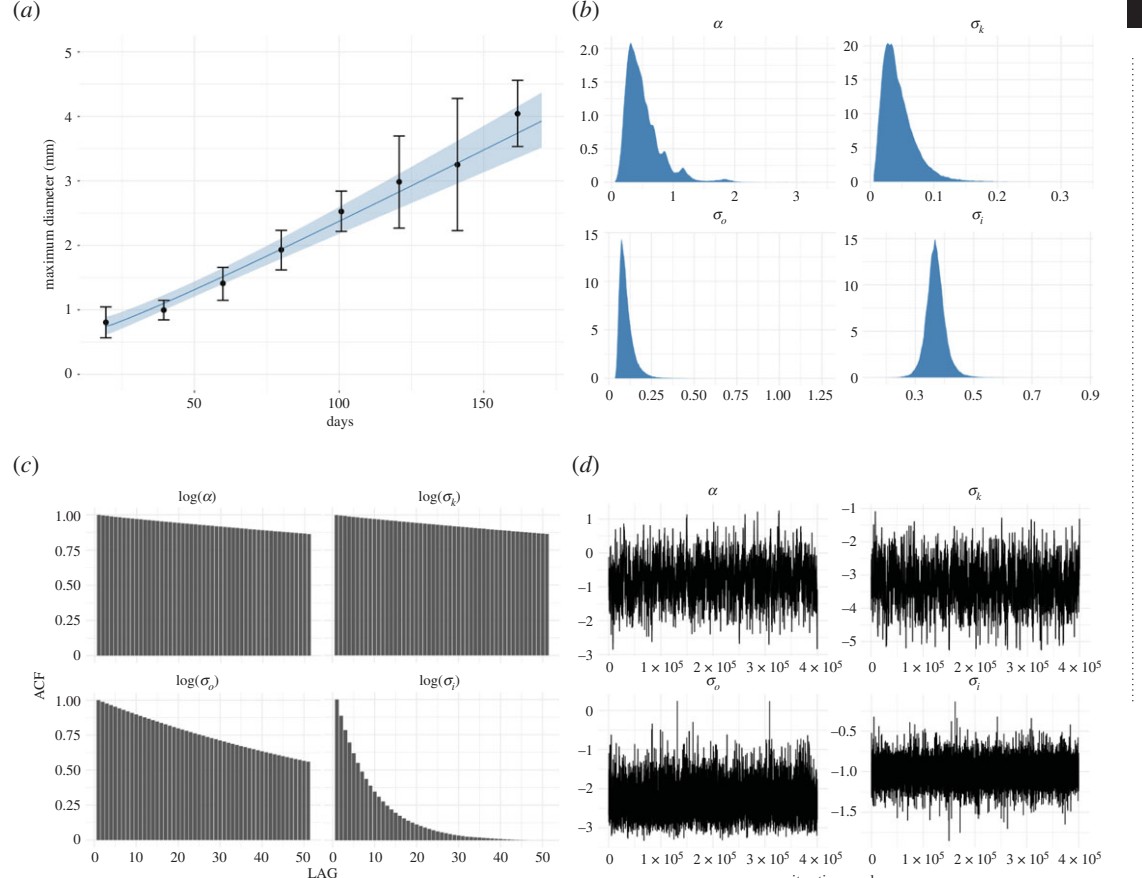

**Figure 3.** The figure shows the results of Bayesian inference for model (2.1) using the data on V-79 cells provided by Folkman & Hochberg ([15], or figure 2b). Plots with (a) label show the predicted mean diameters of spheroids, with black dots standing for measurements, the blue line presenting the diameters predicted by the model, and finally, the shadow area indicating the 95% credibility intervals for the predictions. Plots with (b) labels present the marginal posterior distributions of estimated parameters, whereas plots with labels (c) show the auto-correlation and plots with labels (d) stand for the trace plots of the trajectories of a random walk Metropolis. Simulations performed for $\tilde{\sigma}_i = 1.065 \cdot \sigma_i$ and $q = 13$, cf. (3.9).

In result, we have a vector $\theta = [\alpha, \sigma_k^2, \sigma_o^2, \sigma_i^2]$ of unknown parameters whose coordinates correspond to proliferation rate, kernel size, measurement error and initial colony radius, respectively. For convenience, and to avoid unnecessary constraints, that are $\alpha, \sigma_k, \sigma_o, \sigma_i > 0$, we use logarithms of parameters instead of parameters itself. Finally, we need to define the likelihood function $\ell(D|\theta)$ whose form follows directly from the assumption (3.7) and is given by

$$\ell(D|\theta) = \prod_i \frac{1}{\sqrt{2\pi}\sigma_o} \exp\left(\frac{-\left(\log(r_o^i) - \log(r(t_i))\right)^2}{2\sigma_o^2}\right), \tag{3.10}$$

where $r(t_i)$ is the colony radius at time $t_i$ for vector of parameters $\theta$.

To sample the parameter space $\Theta$, we choose a random walk Metropolis algorithm, i.e. the proposal distribution is a multivariate normal distribution

$$\tilde{\theta} = \theta_j + Z, \quad \text{where } Z \sim N(0, s \cdot \text{Id}), \tag{3.11}$$

where $s$ is a step-size, tuned in a way that the acceptance probability of the candidate $\tilde{\theta}$ is close to the optimal one [22].

Finally, to complete the description of the model, we need to provide the specific a priori distribution $\pi(\theta)$. We assume the time scale of simulations corresponding to the time scale of the experiments conducted by Folkman & Hochberg [15], therefore, we set the time unit to 1 day. For similar reasons, we take 1 mm as the length unit. We assume that a priori distributions for all unknown parameters are independent lognormal distributions. When determining the a priori distributions of proliferation parameters for particular cell lines, we use the mean doubling time for each cell type, see table 1, and

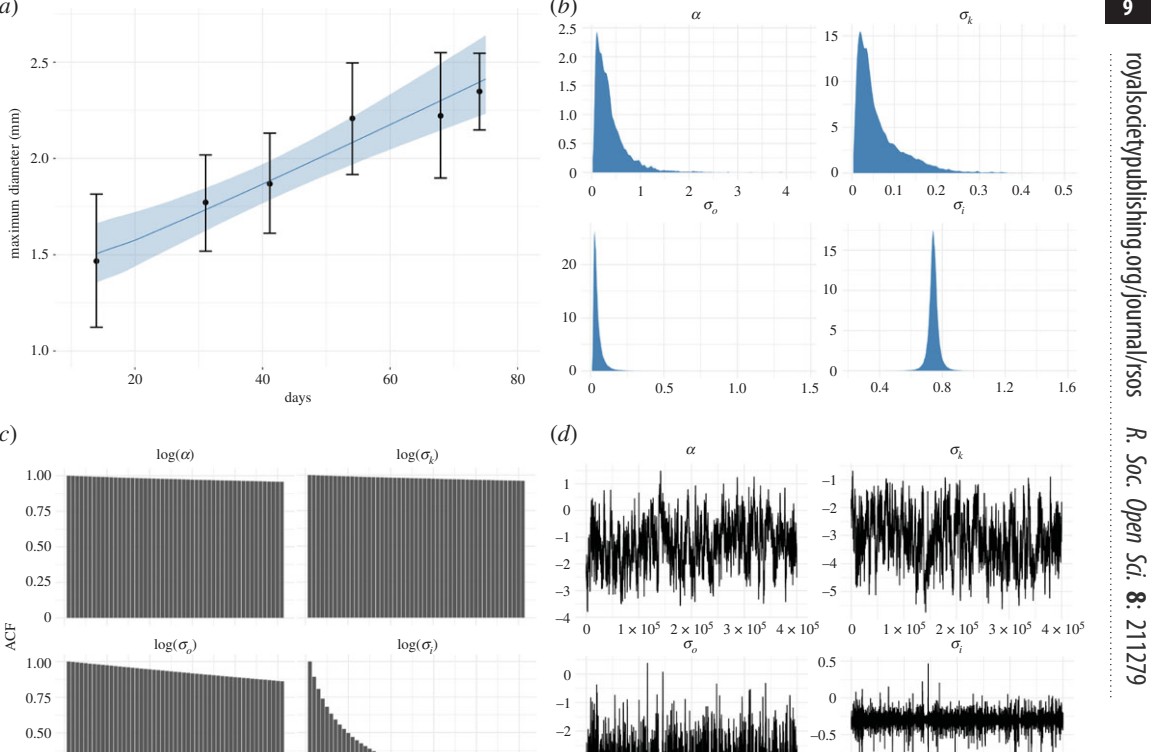

**Figure 4.** The figure shows the results of Bayesian inference for model (2.1) using the data on B-16 melanoma cells provided by Folkman & Hochberg ([15], or figure 2c). Plot with (a) label shows the predicted mean diameters of spheroids, with black dots standing for measurements, the blue line presenting the diameters predicted by the model, and finally, the shadow area indicating the 95% credibility intervals for the predictions. Plot with (b) label presents the marginal posterior distributions of estimated parameters, whereas plot with (c) label shows the auto-correlation, and plot with (d) label stands for the trace plots of the trajectories of a random walk Metropolis. Simulations performed for $\tilde{\sigma}_i = 1.06 \cdot \sigma_i$ and $q = 13$, cf. (3.9).

**Table 1.** Division times, mean radius of the cell, and radius of the initial colonies for three considered cell lines.

|  | L-5178Y | V-79 | B-16 |
|---|---|---|---|
| doubling time | ∼11.3 h [23] | ∼16 h [24,25] | ∼18 h [26] |
| cell diameter | 10 − 20 μm [27] | 10 μm [28] | 15.4 μm [29] |
| initial colony radius | 0.264 mm [15] | 0.403 mm [15] | 0.733 mm [15] |

we assume that cells in a colony proliferate 2.4 times slower. Within this framework, we get the 'a priori' proliferation rates equal to 1.4, 1.04, 0.9, for L-5178Y, V-79 and B-16 cell lines, respectively. According to Folkman & Hochberg [15], the proliferating ring is restricted to the outer layer of several cells. Therefore we assume that kernel size is equal to six times the cell diameter, which gives us mean values of kernel size equal to 0.06 for L-5178Y and V-79 and 0.09 for B-16 cell lines. We assume that the mean value of *a priori* distributions of initial colony radius is equal to the mean radius of the first measurements; moreover, the false and positive errors are equally probable, therefore we set the mean values of measurement errors equal to zero. For $\alpha$, $\sigma_k$ and $\sigma_i$, we set the standard deviation to 1, whereas for $\sigma_o$ we set it equal to 5 to cover the rather higher uncertainty of observation error over other parameters.

# 4. Stability of posterior distribution

Bearing in mind that we approximate the posterior distribution (3.1) using numerical solutions rather than exact ones, we need to prove that our approximation is indeed close to the actual one. At first, in

§4.1, we propose a regularization of the quantile function used for colony radius approximation (3.5) and prove its Lipschitz continuity. We refer the interested reader to appendix A for the detailed formulation of an auxiliary lemma, which is needed to show that the inverse of the cumulative distribution function satisfies the continuity estimate. Within appendix B, we also recall the necessary notions from measure theory and we formulate theorems about the existence and uniqueness of measure solutions. In §4.2, we prove the stability of posterior probability distribution of $\theta$ with respect to the EBT approximation of (2.3).

## 4.1. Regularization of percentile function and its Lipschitz continuity

The quantile function (3.5) used to determine the colony radius at time $t$ is not invertible, which is crucial for further analysis as we retrieve radius from the measure solution to obtain the likelihood function (3.10). Hence, we propose the following regularization. Consider $\mu \in \mathcal{M}^+(\mathbb{R}^+)$ and continuous function $\eta : \mathbb{R} \to \mathbb{R}^+$. We define

$$\mathcal{F}^\eta(x, \mu) = \frac{\int_{-\infty}^x \mu * \eta(y) \, dy}{\|\mu\|_{TV}}, \tag{4.1}$$

where $\mu * \eta$ is a convolution of the measure $\mu$ with the function $\eta$, which we call a regularizing kernel, and $\|\mu\|_{TV}$ is the total variation norm defined by (B2). We note that such a convolution is a function as well. We impose the additional assumptions on $\eta$ and initial condition $\mu_0$ that guarantee stability properties of (4.1).

**Assumption 4.1. (On initial condition $\mu_0$ and regularizing kernel $\eta$).**
We assume that:

(A) There exist $\kappa < 0.05$ and $\varepsilon(\kappa) > 0$ such that

$$\mu_0 * \eta(y) > \varepsilon(\kappa) > 0 \tag{4.2}$$

for all $y$ such that $0 < \kappa < \mathcal{F}^\eta(y, \mu_0) < 1 - \kappa$.

(B) $1/\varepsilon(\mu_0) \geq \|\mu_0\|_{TV} \geq \varepsilon(\mu_0) > 0$ for some $\varepsilon(\mu_0)$.

(C) $\eta$ is a non-negative smooth function such that $\int_{\mathbb{R}^+} \eta(y) \, dy = 1$ and $\eta$ is bounded, i.e. $\eta \in L^\infty$. Moreover, for some small $\varepsilon(\eta)$ we have $\eta(y) = 0$ for $|y| \geq \varepsilon(\eta)$ and $\eta(y) = 1$ for $|y| \leq \varepsilon(\eta)/2$.

**Remark 4.2.** Assumption $\eta(y) = 0$ for $|y| \geq \varepsilon(\eta)$ is purely technical and can be relaxed to the sufficiently fast decaying distributions like normal distribution or Laplace distribution, which is used in this paper, cf. (3.6).

**Remark 4.3.** The assumption 4.1 is usually satisfied. In particular, (A) is natural and is preserved for measures being approximated by particles.

Let $\mu_0 \in \mathcal{M}^+(\mathbb{R}^+)$ be a measure with density $p(r)$ which has a connected support in $\mathbb{R}^+$. Then, measure $\mu_0 * \eta$ has slightly larger connected support of the form $(-\delta, P)$ and the map $y \mapsto \mathcal{F}^\eta(y, \mu_0)$ is strictly increasing. Moreover, condition $\kappa < \mathcal{F}^\eta(y, \mu_0) < 1 - \kappa$ is satisfied if and only if $y \in (-\tilde{\delta}, \tilde{P}) \subset (-\delta, P)$. Hence, to fulfil (4.2) we may choose

$$\varepsilon(\kappa) := \inf_{y \in (-\tilde{\delta}, \tilde{P})} \mu * \eta(x),$$

which is strictly positive by connectness of the support.

Now, we prove that (4.2) is preserved under particle approximation of initial condition with uniform constants assuming that the discretization is sufficiently small. Consider measure on $[0, R_0]$ defined with

$$\mu_0^N = \sum_{i=1}^N \delta_{x_i^N} \int_{x_{i-1}^N}^{x_i^N} p(r) \, dr, \quad x_i^N = \frac{i}{N} R_0, \quad i = 1, \dots, N. \tag{4.3}$$

Then, if $\eta(y) \neq 0$ for $|y| \leq \varepsilon(\eta)$ and $\eta(y) = 1$ for $|y| \leq \varepsilon(\eta)/2$, we have

$$\mu_0^N * \eta(y) = \sum_{i:\,|x_i^N - y| \leq \varepsilon(\eta)} \eta(y - x_i^N) \int_{x_{i-1}^N}^{x_i^N} p(r)\,\mathrm{d}r \geq \sum_{i:\,|x_i^N - y| \leq \varepsilon(\eta)/2} \eta(y - x_i^N) \int_{x_{i-1}^N}^{x_i^N} p(r)\,\mathrm{d}r$$

$$= \sum_{i:\,|x_i^N - y| \leq \varepsilon(\eta)/2} \int_{x_{i-1}^N}^{x_i^N} p(r)\,\mathrm{d}r \geq \int_{y-\varepsilon(\eta)/2-R_0/N}^{y+\varepsilon(\eta)/2+R_0/N} p(r)\,\mathrm{d}r \geq \int_{y-\varepsilon(\eta)/4}^{y+\varepsilon(\eta)/4} p(r)\,\mathrm{d}r,$$

where in the last line we assumed additionally that $R_0/N \leq \frac{\varepsilon(\eta)}{4}$, i.e. discretization is sufficiently small. Now, it is enough to apply reasoning from the first part of the remark to the measure

$$y \mapsto \int_{y-\varepsilon(\eta)/4}^{y+\varepsilon(\eta)/4} p(r)\,\mathrm{d}r.$$

We choose $\kappa < 0.05$ so that the function $\mathcal{F}^\eta(x, \mu)$ is invertible around 0.95 which corresponds to our quantile function, see (3.5). Moreover, since $\int_{\mathbb{R}^+} \eta(y)\,\mathrm{d}y = 1$ we have

$$\|\mu\|_{\mathrm{TV}} = \|\mu * \eta\|_{\mathrm{TV}},$$

and consequently $\mathcal{F}^\eta(x, \mu) \in [0, 1]$.

Now, we prove that property (4.2) propagates with time, up to an exponential constant.

**Lemma 4.4.** *Let* $\mu_t \in \mathcal{M}^+(\mathbb{R}^+)$ *be a measure solution to* (2.3) *with initial condition* $\mu_0$ *and let* $\mu_t^N$ *be the particle approximation defined in* (2.6)–(2.7). *Then, there is a constant* $C$ *depending continuously on parameters and the size of initial conditions* $\|\mu_0\|_{\mathrm{TV}}$ *such that*

and
$$\left.\begin{array}{ll} \mu_t * \eta(y) \geq \varepsilon(\kappa) \cdot e^{-Ct} > 0, & \|\mu_t\|_{\mathrm{TV}} \geq \varepsilon(\mu_0) \cdot e^{-Ct} > 0 \\ \mu_t^N * \eta(y) \geq \varepsilon(\kappa) \cdot e^{-Ct} > 0, & \|\mu_t^N\|_{\mathrm{TV}} \geq \varepsilon(\mu_0) \cdot e^{-Ct} > 0. \end{array}\right\} \tag{4.4}$$

*Proof.* Measure solutions to (2.3) are non-negative and uniformly bounded with respect to $\|\cdot\|_{\mathrm{TV}}$, with a constant depending only on the initial condition, time and parameters, see theorem B.1 in appendix B. The proof of that theorem can be found in our companion paper [6]. Using (2.3), we deduce

$$\partial_t \mu_t(R) \geq -\mu_t(R) \int_0^\infty L(R, r)\,\mathrm{d}\mu_t(r) \geq -\mu_t(R)\,\|L\|_\infty\,\|\mu_t\|_{\mathrm{TV}} \geq -C\,\mu_t(R)$$

understood in the sense of distributions. Taking convolution with $\eta$, we deduce

$$\partial_t \mu_t * \eta(R) \geq -C\,\mu_t * \eta(R),$$

which implies

$$\partial_t[\mu_t * \eta(R)\,e^{Ct}] \geq 0.$$

Integrating in time we conclude estimates for $\mu_t$. To establish estimates for $\mu_t^N$, we observe that (2.6) implies distributional inequality

$$\partial_t \mu_t^N(R) \geq -\mu_t^N(R) \int_0^\infty L(R, r)\,\mathrm{d}\mu_t^N(r),$$

so that the proof above applies also to $\mu_t^N$. As $\|\mu_0\|_{\mathrm{TV}} = \|\mu_0^N\|_{\mathrm{TV}}$, the proof is concluded. ∎

Now, we are in position to prove that on the appropriate set the function $\mathcal{F}^\eta$ satisfies lemma A.1. For simplicity, we denote by $\hat\theta = [\alpha, \sigma_k, \sigma_i]$. Note that the constants in the estimates (4.4) are independent of $\hat\theta$ assuming that $\alpha, \sigma_k, \sigma_i$ are in the certain range of values, that is usually bounded and separated from zero. For the forthcoming consideration, it is convenient to define two sets $R$ and $S$ consisting of solutions to equation (2.3) and the numerical schemes (2.6)–(2.7), respectively. Moreover, to investigate stability

properties of radial solutions we use weighted flat norm defined by (B4).

$$R = \{\mu_t \in \mathcal{M}^+(\mathbb{R}^+) : \mu_t \text{ is a solution to } (2.3), \text{ for } 0 \leq t \leq T,$$

$$\text{with } \hat{\theta} \in \left[ \varepsilon(\hat{\theta}), \frac{1}{\varepsilon(\hat{\theta})} \right], \text{ and initial}$$

condition $\mu_0$ satisfying assumption 4.1 with the same constant $\varepsilon(\kappa)$ and $\varepsilon(\mu_0)\}$,

$$S = \{\mu_t^N \in \mathcal{M}^+(\mathbb{R}^+) : \mu_t^N \text{ is a solution to } (2.6\text{--}2.7), \text{ for } 0 \leq t \leq T$$

$$\text{with } \hat{\theta} \in \left[ \varepsilon(\hat{\theta}), \frac{1}{\varepsilon(\hat{\theta})} \right], \text{ and initial condition } \mu_0 \text{ satisfying assumption 4.1}$$

with the same constant $\varepsilon(\kappa)$ and $\varepsilon(\mu_0)\}$.

**Theorem 4.5.** *Let $\kappa$ be a small number from assumption 4.1. Then, there are $0 < a_\kappa < b_\kappa$ such that for all $t \in [0, T]$, the function*

$$\mathcal{F}^\eta(x, \mu) : (a_\kappa, b_\kappa) \times (R \cup S) \to (\kappa, 1 - \kappa),$$

*satisfies lemma A.1. Hence, we can define $\mathcal{G}_\mu := x$ as the unique solution of equation $\mathcal{F}^\eta(x, \mu) = 0.95$ where $\mu$ is fixed. Moreover, for $R_0$ all such that $\sup_{t \in [0,T]} \int_{(R_0, \infty)} \mathrm{d}\mu_t < 0.05$ we have*

$$|\mathcal{G}_{\mu_t} - \mathcal{G}_{\nu_t}| \leq C \left[ (2R_0 + 1) \left\| \frac{\mu_t - \nu_t}{r} \right\|_{BL^*} + e^{-R_0/2} \right], \tag{4.5}$$

*for some constant $C$ depending continuously on $\varepsilon(\mu_0)$, $\varepsilon(\hat{\theta})$, $\varepsilon(\kappa)$, $\kappa$, $a_\kappa$, $b_\kappa$.*

**Remark 4.6.** The existence of an appropriate $R_0$ follows from remark B.3.

*Proof.* First, we note that $\partial_x \mathcal{F}^\eta(x, \mu_t) = \mu_t * \eta \geq e^{-CT} \mu_0 * \eta \geq \varepsilon_\kappa$ so that we can always find such $a_\kappa$ uniformly for all elements of $R$. Existence of such $b_\kappa$ follows from uniform tail estimate (B6) in theorems B.1 and B.5.

Concerning lemma A.1, condition (A1) is satisfied. For the second one, we write

$$\mathcal{F}^\eta(x, \mu_t) - \mathcal{F}^\eta(x, \nu_t) = \frac{\int_{-\infty}^x \mu_t * \eta(y) \, \mathrm{d}y}{\|\mu_t\|_{\mathrm{TV}}} - \frac{\int_{-\infty}^x \nu_t * \eta(y) \, \mathrm{d}y}{\|\nu_t\|_{\mathrm{TV}}}$$

$$\leq \frac{\int_{-\infty}^x (\mu_t - \nu_t) * \eta(y) \, \mathrm{d}y}{\|\mu_t\|_{\mathrm{TV}}} + \int_{-\infty}^x \nu_t * \eta(y) \, \mathrm{d}y \left( \frac{1}{\|\mu_t\|_{\mathrm{TV}}} - \frac{1}{\|\nu_t\|_{\mathrm{TV}}} \right) =: A + B.$$

Note that

$$\int_{-\infty}^x (\mu_t - \nu_t) * \eta(y) \, \mathrm{d}y = \int_{-\infty}^x \int_{\mathbb{R}^+} \eta(y - z) \mathrm{d}(\mu_t - \nu_t)(z) \, \mathrm{d}y = \int_{\mathbb{R}^+} \int_{-\infty}^x \eta(y - z) \, \mathrm{d}y \mathrm{d}(\mu_t - \nu_t)(z).$$

The function $z \mapsto \int_{-\infty}^x \eta(y - z) \, \mathrm{d}y$ is bounded by an $L^1$ norm of $\eta$ and Lipschitz continuous as for all $z_1, z_2$ we have

$$\left| \int_{-\infty}^x \eta(y - z_1) \, \mathrm{d}y - \int_{-\infty}^x \eta(y - z_2) \, \mathrm{d}y \right| \leq C_{Lip}(\eta) \, b_\kappa \, |z_1 - z_2|,$$

and $\eta$ was assumed to be Lipschitz continuous with constant $C_{Lip}(\eta)$. It follows that

$$\int_{-\infty}^x (\mu_t - \nu_t) * \eta(y) \, \mathrm{d}y \leq (\|\eta\|_1 + C_{Lip}(\eta) \, b_\kappa) \|\mu_t - \nu_t\|_{BL^*}$$

and consequently, using lemma 4.4, we can estimate term $A$ with

$$A \leq \frac{(\|\eta\|_1 + C_{Lip}(\eta) \, b_\kappa) \, e^{CT}}{\varepsilon(\mu_0)} \|\mu_t - \nu_t\|_{BL^*}.$$

For term $B$, we observe $\|\mu_t\|_{\mathrm{TV}} = \int_{\mathbb{R}^+} \mathrm{d}\mu_t$ and $\|\nu_t\|_{\mathrm{TV}} = \int_{\mathbb{R}^+} \mathrm{d}\nu_t$ so that

$$\left( \frac{1}{\|\mu_t\|_{\mathrm{TV}}} - \frac{1}{\|\nu_t\|_{\mathrm{TV}}} \right) = \frac{1}{\|\mu_t\|_{\mathrm{TV}} \|\nu_t\|_{\mathrm{TV}}} \int_{\mathbb{R}^+} \mathrm{d}(\mu_t - \nu_t) \leq \frac{e^{2CT}}{\varepsilon(\mu_0)^2} \|\mu_t - \nu_t\|_{BL^*},$$

where $C$ comes from lemma 4.4. By virtue of Young's convolutional inequality, we observe that

$$\left| \int_{-\infty}^{x} \nu_t * \eta(y) \, dy \right| \leq \left| \int_{\mathbb{R}^+} \nu_t * \eta(y) \, dy \right| \leq \|\eta\|_1 \|\nu_t\|_{\mathrm{TV}},$$

which implies

$$B \leq \|\eta\|_1 \|\nu_t\|_{\mathrm{TV}} \frac{e^{2CT}}{\varepsilon(\mu_0)^2} \|\mu_t - \nu_t\|_{BL^*}.$$

Finally, we note that for all $R_0 > 0$ we have interpolation inequality

$$\|\mu_t - \nu_t\|_{BL^*} \leq (2R_0 + 1) \left\| \frac{\mu_t - \nu_t}{r} \right\|_{BL^*} + C\, e^{-R_0/2}. \tag{4.6}$$

Indeed, for all $\psi \in BL(\mathbb{R}^+)$ with $\|\psi\|_{BL} \leq 1$

$$\left| \int_{\mathbb{R}^+} \psi(r) \mathrm{d}(\mu_t - \nu_t)(r) \right| = \left| \int_{r \leq R_0} \frac{r\psi(r)}{r} \, \mathrm{d}(\mu_t - \nu_t)(r) \right| + \left| \int_{r > R_0} \frac{r\psi(r)}{r} \, \mathrm{d}\mu_t(r) \right| + \left| \int_{r > R_0} \frac{r\psi(r)}{r} \, \mathrm{d}\nu_t(r) \right|.$$

For the first term, we note that the map $[0, R_0] \ni r \mapsto r\psi(r)$ is bounded with $R_0$ and Lipschitz continuous with constant $(1 + R_0)$. Hence,

$$\left| \int_{r \leq R_0} \frac{r\psi(r)}{r} \, \mathrm{d}(\mu_t - \nu_t)(r) \right| \leq (1 + 2R_0) \left\| \frac{\mu_t - \nu_t}{r} \right\|_{BL^*[0, R_0]}.$$

For the second and third term, we use propagation of moments estimate (B 5). Indeed,

$$\left| \int_{r > R_0} \frac{r\psi(r)}{r} \, \mathrm{d}\mu_t(r) \right| \leq \int_{r > R_0} \frac{e^{r/2}}{e^{R_0/2}} \, \mathrm{d}\mu_t(r) \leq e^{-R_0/2} \int_{\mathbb{R}^+} e^{r/2} \, \mathrm{d}\mu_t(r) \leq C\, e^{-R_0/2}.$$

Taking supremum over all $\psi \in BL(\mathbb{R}^+)$ with $\|\psi\|_{BL} \leq 1$, we conclude the proof of (4.6) which proves

$$|\mathcal{F}^\eta(x, \mu_t) - \mathcal{F}^\eta(x, \nu_t)| \leq C \left[ (2R_0 + 1) \left\| \frac{\mu_t - \nu_t}{r} \right\|_{BL^*} + e^{-R_0/2} \right],$$

where $C$ may depend on $\varepsilon(\mu_0)$, $\varepsilon(\hat{\theta})$, $\varepsilon(\kappa)$, $\kappa$, $a_\kappa$, $b_\kappa$. Now, as $\kappa < 0.05$, we obtain (4.5) directly from lemma A.1. ∎

## 4.2. Proof of stability of posterior distribution

Let us remind that posterior distribution is given by (3.1) with likelihood function defined by (3.10). The actual colony radii $r(t_i)$ are the function of the solution, i.e. $r(t_i) = \mathcal{G}_{\mu_{t_i}}$. Hence, we may write

$$\ell(D|\theta)^\mu = \prod_{i=1}^{l} \frac{1}{\sqrt{2\pi}\sigma_o} \exp\left( \frac{-\left( \log(r_o^i) - \log(\mathcal{G}_{\mu_{t_i}(\hat{\theta})}) \right)^2}{2\sigma_o^2} \right), \tag{4.7}$$

where we added a superscript $\mu$ to denote dependence on the measure solution $\mu$. Recall that as in theorem 4.5, we work in the set $R \cup S$ of measure solutions obtained with appropriate initial conditions and values of parameters as well as solutions to the numerical scheme.

**Lemma 4.7.** *Let $\varepsilon(D)$ be such that $1/\varepsilon(D) \geq r_0^i \geq \varepsilon(D)$. Then, there exists $\varepsilon(\theta, D) > 0$ such that for all $\mu_t \in R$ as in theorem 4.5, we have*

$$\ell(D|\theta)^\mu \geq \varepsilon(\theta, D), \quad \int_\Theta \ell(D|\theta)^\mu \pi(\theta) \geq \varepsilon(\theta, D).$$

**Remark 4.8.** The existence of $\varepsilon(D)$ such as in lemma 4.7 is due to the nature of the data.

*Proof.* Note that $\log a_\kappa \leq \log(\mathcal{G}_{\mu_{t_i}(\hat{\theta})}) \leq \log b_\kappa$, so the first inequality follows directly from assumptions and formula (4.7), while the second one follows from the first after noting that $\int_\Theta \pi(\theta) = 1$. ∎

**Lemma 4.9.** *Let $0 < a < b$, $\zeta > 0$ and $w \in \mathbb{R}$. Then, function*

$$(a, b) \ni y \mapsto \mathcal{F}(y) := \exp\left( -\frac{(w - \log(y))^2}{\zeta} \right),$$

*is Lipschitz continuous with constant* $2\,((w + \log\,(b))/\zeta\,a)$.

*Proof.* Clearly, the function $x \mapsto e^x$ for $x \leq 0$ is 1-Lipschitz. Moreover

$$\left| \partial_y \frac{(w - \log(y))^2}{\zeta} \right| = \left| -2 \frac{(w - \log(y))}{\zeta} \frac{1}{y} \right| \leq 2 \frac{w + \log(b)}{\zeta\,a}.$$

The conclusion follows. ∎

**Theorem 4.10 (Lipschitz continuity of posterior distributions).** *Let $\pi_1$, $\pi_2$ be a posteriori distributions computed using measure solutions $\mu_t$, $\nu_t \in R$, i.e.*

$$\pi_1(\theta|D) = \frac{\ell(D|\theta)^\mu \pi(\theta)}{\int_\Theta \ell(D|\theta)^\mu \pi(\theta)}, \quad \pi_2(\theta|D) = \frac{\ell(D|\theta)^\nu \pi(\theta)}{\int_\Theta \ell(D|\theta)^\nu \pi(\theta)}.$$

*Assume additionally that $\sigma_o \geq \varepsilon(\hat{\theta}) > 0$. Then, there is a constant $C$ such that for all $R_0 > 0$,*

$$\|\pi_1(\theta|D) - \pi_2(\theta|D)\|_{\mathrm{TV}} \leq C\left[ (2R_0 + 1) \left\| \frac{\mu_t - \nu_t}{r} \right\|_{BL^*} + e^{-R_0/2} \right]. \tag{4.8}$$

*Proof.* First, we observe that

$$\|\pi_1(\theta|D) - \pi_2(\theta|D)\|_{\mathrm{TV}}$$

$$\leq \frac{\int_\Theta |\ell(D|\theta)^\mu - \ell(D|\theta)^\nu|\,\pi(\theta)}{\int_\Theta \ell(D|\theta)^\mu \pi(\theta)} + \int_\Theta \ell(D|\theta)^\nu\,\pi(\theta) \frac{\int_\Theta |\ell(D|\theta)^\mu - \ell(D|\theta)^\nu|\,\pi(\theta)}{\int_\Theta \ell(D|\theta)^\mu \pi(\theta)\,\int_\Theta \ell(D|\theta)^\nu \pi(\theta)}$$

$$\leq \frac{\int_\Theta |\ell(D|\theta)^\mu - \ell(D|\theta)^\nu|\,\pi(\theta)}{\int_\Theta \ell(D|\theta)^\mu \pi(\theta)} + \frac{\int_\Theta |\ell(D|\theta)^\mu - \ell(D|\theta)^\nu|\,\pi(\theta)}{\int_\Theta \ell(D|\theta)^\mu \pi(\theta)}. \tag{4.9}$$

We note that triangle inequality and inequality $0 \leq e^{-x} \leq 1$ for $x \leq 0$ implies

$$|\ell(D|\theta)^\mu - \ell(D|\theta)^\nu|$$

$$\leq \frac{1}{\sqrt{2\pi}\sigma_o} \sum_{i=1}^l \left| \exp\left( \frac{-\left( \log(r_o^i) - \log(\mathcal{G}_{\mu_{t_i}(\hat{\theta})}) \right)^2}{2\sigma_o^2} \right) - \exp\left( \frac{-\left( \log(r_o^i) - \log(\mathcal{G}_{\mu_{t_i}(\hat{\theta})}) \right)^2}{2\sigma_o^2} \right) \right|.$$

Then from lemma 4.9, we obtain that

$$|\ell(D|\theta)^\mu - \ell(D|\theta)^\nu| \leq \frac{1}{\sqrt{2\pi}\sigma_o} \sum_{i=1}^l 2 \frac{|\log(r_o^i)| + \log(b_\kappa)}{\sigma_o\,a_\kappa} |\mathcal{G}_{\mu_{t_i}(\hat{\theta})} - \mathcal{G}_{\nu_{t_i}(\hat{\theta})}|,$$

where $a_\kappa$ and $b_\kappa$ are such that $0 < a_\kappa \leq \mathcal{G}_{\mu_{t_i}(\hat{\theta})}, \mathcal{G}_{\nu_{t_i}(\hat{\theta})} \leq b_\kappa$. Letting

$$C_1 := \frac{l}{\sqrt{2\pi}\,\varepsilon(\theta, D)} \frac{\sup_{i=1}^l |\log(r_o^i)| + \log(b)}{\varepsilon(\theta, D)\,a}$$

we obtain

$$|\ell(D|\theta)^\mu - \ell(D|\theta)^\nu| \leq C_1 \sup_{1 \leq i \leq l} |\mathcal{G}_{\mu_{t_i}(\hat{\theta})} - \mathcal{G}_{\nu_{t_i}(\hat{\theta})}|.$$

Then, equation (4.5) implies

$$|\ell(D|\theta)^\mu - \ell(D|\theta)^\nu| \leq C\left[ (2R_0 + 1) \left\| \frac{\mu_t - \nu_t}{r} \right\|_{BL^*} + e^{-R_0/2} \right]$$

for a possibly larger constant $C$. Hence, from (4.9) we deduce

$$\|\pi_1(\theta|D) - \pi_2(\theta|D)\|_{\mathrm{TV}} \leq \frac{2C}{\varepsilon(\theta, D)} \left[ (2R_0 + 1) \left\| \frac{\mu_t - \nu_t}{r} \right\|_{BL^*} + e^{-R_0/2} \right]$$

where we applied lemma 4.7 and $\int_\Theta \pi(\theta) = 1$. ∎

**Theorem 4.11 (Stability of posterior distribution with respect to particle approximation).** *Let $\mu_t \in R$ be a measure solution to (2.3) with initial condition $\mu_0$. Let $\mu_t^N = m_i(t)\,\delta_{x_i}$ where $m_i(t)$ solves the system of ODEs*

*(2.6) and $m_i(0)$ are chosen as in (2.7). Then, if we let*

$$\pi(\theta|D) = \frac{\ell(D|\theta)^{\mu}\pi(\theta)}{\int_{\Theta}\ell(D|\theta)^{\mu}\pi(\theta)} \quad \text{and} \quad \pi_N(\theta|D) = \frac{\ell(D|\theta)^{\mu^N}\pi(\theta)}{\int_{\Theta}\ell(D|\theta)^{\mu^N}\pi(\theta)}.$$

*we have*

$$\|\pi(\theta|D) - \pi_N(\theta|D)\|_{\mathrm{TV}} \leq C\left[(2R_0+1)\left(\frac{R_0^2}{N} + \mathrm{e}^{-R_0}\right) + \mathrm{e}^{-R_0/2}\right].$$

*In particular,*

$$\lim_{R_0 \to \infty}\lim_{N \to \infty}\|\pi(\theta|D) - \pi_N(\theta|D)\|_{\mathrm{TV}} = 0.$$

*Proof.* We let $\nu(t) = \mu^N(t)$ in (4.8) and use (B 7) to conclude the proof. ∎

# 5. Simulations results

To perform statistical inference, i.e. to predict the growth curve of diameters of multicellular spheroids and to identify parameters related to proliferation rate, kernel size, measurement error and initial colony radius we use our Bayesian model with MAP estimator. We run 450 000 iterations of the Metropolis–Hastings algorithm, and we discarded the first 50 000 iteration as a burn-in. The pseudo-code for this algorithm can be found in appendix C. Figures 2*a*, 3*a* and 4*a* present the predicted growth curves of diameters of multicellular spheroids (blue line) together with experimental measurements (the black dots) and 95% credible intervals for the prediction (light blue shadowed area) for L-5178Y cells, V-79 cells and B-16 melanoma cells, respectively. Figures 2*b*, 3*b* and 4*b* present marginal posterior densities of parameters of interest for appropriate cell lines, whereas figures 2*c*, 3*c* and 4*c* stand for auto-correlation plots. Finally, figures 2*d*, 3*d* and 4*d* show the trace plots of the trajectories of algorithm 1. For all simulations, we adjust the step size $s$ of proposal distribution from (3.11) to achieve optimal acceptance ratio approximately 23%. Data and relevant code for this research work are stored in GitHub: https://github.com/Zuzanna-Szymanska/Non-local-proliferation-model and have been archived within the Zenodo repository: https://doi.org/10.5281/zenodo.5565314 [30].

The Bayesian prediction proves to be very accurate for the prognosis of dynamics of diameters of multicellular spheroids. However, for practical purposes, namely quantitative modelling of cancer growth the MAP estimator seems to be more accurate. Using the MAP estimator we obtain the prediction of diameters dynamics very close to the Bayesian one; however, this approach allows us to obtain more accurate parameters for the proposed proliferation function (2.1). Figure 5 presents the predicted dynamics of diameters for all considered datasets, whose predictions were obtained using the MAP estimator (red curve) and the Bayesian estimator (blue line). Using the MAP estimator, we obtained $\alpha = 1.7264$, $\sigma_k = 0.0806$, $\sigma_o = 0.0957$ and $\sigma_i = 0.2469$ for the mouse lymphoma L-5178Y cells, see figure 5*a*. Adopting the same estimator, we get $\alpha = 0.3603$, $\sigma_k = 0.0479$, $\sigma_o = 0.0649$ and $\sigma_i = 0.3744$ for the Chinese hamster lung cell line V-79, see figure 5*b*. Finally, for B-16 melanoma cell line, we get the MAP estimator $\alpha = 0.3616$, $\sigma_k = 0.0342$, $\sigma_o = 0.0256$ and $\sigma_i = 0.7518$, see figure 5*c*.

While analysing the trace plots and auto-correlation plots, it becomes noticeable that the algorithm converges with different speeds along different dimensions of parameter space. Moreover, we see that for L-5178Y and V-79 cells (datasets a and b) MCMC algorithm mixes rather well, while the convergence of the algorithm for B-16 cells (dataset c) is significantly slower. Perhaps this is due to the correlation between parameters $\alpha$ and $\sigma_k$ that for the cell line whose linear growth is the slowest becomes more apparent. We speculate, that for such challenging cases, it might be worth trying more sophisticated algorithms Metropolis–Hastings MCMC; however, the issue goes beyond the scope of the current paper, the main aim of which was to propose a new proliferation function suitable to incorporate into models describing solid tumour dynamics. Finally, we observe that estimated parameters in all examples are similar, which bolsters the surmise that the model does not overfit the data. Therefore, our model provides a quite good approximation of reality in the considered time window.

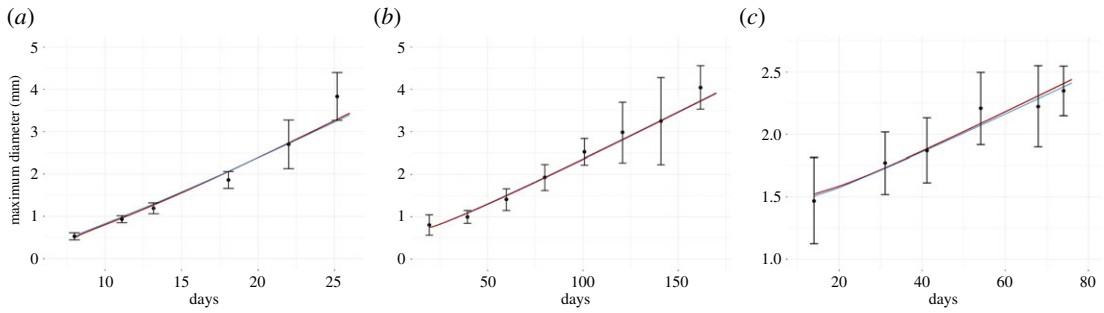

**Figure 5.** Predictions of growth curves of diameters of spheroids obtained via MAP estimator—drawn on diagrams in red. Plot with (a) label shows the prognosis for mouse lymphoma L-5178Y cells obtained for $\alpha = 1.7264$, $\sigma_k = 0.0806$, $\sigma_o = 0.0957$ and $\sigma_i = 0.2469$. Plot with (b) label presents the prediction for the V-79 Chinese hamster lung cell line obtained for parameters $\alpha = 0.3603$, $\sigma_k = 0.0479$, $\sigma_o = 0.0649$ and $\sigma_i = 0.3744$. Plot with (c) label shows the best fit of predicted growth curve of B-16 melanoma cell line obtained for $\alpha = 0.3616$, $\sigma_k = 0.0342$, $\sigma_o = 0.0256$ and $\sigma_i = 0.7518$. For comparison, the blue lines stand for appropriates the Bayesian predictions redrawn from figures 2a, 3a and 4a. In all plots, both lines almost overlap. (a) L-5178Y cell line, (b) V-79 cell line and (c) B-16 melanoma cell line.

# 6. Conclusion

In this paper, we propose a non-local function (2.1) to describe the proliferation dynamics of cells living within a colony whose growth is restricted to the outer layer of several viable individuals. To estimate the range of applicability of the model, we refer to the experimental data on cancer multicellular spheroids growth published by Folkman & Hochberg [15]. To deal with the low regularity of the kernel given by (2.2) as well as to improve solution accuracy and algorithm performance, we reformulate the initial model using spherical coordinates (2.3). Then, we performed parameter estimation of the model given by (2.3) based on three datasets on the dynamics of multicellular spheroids growth in three-dimensional culture with medium being replenished and open space being available. For all considered datasets, we observe that the dynamics of the colonies' growths predicted by our model are quite accurate.

An interesting question arises, to what extent the proposed description of cell proliferation is suitable to incorporate into more complex cancer models. Whether the introduction of the proposed non-local proliferation function will bring more accurate quantitative predictions of solid tumour growth or not? To answer that question, it seems interesting to relate the estimated kernel radii to the distance that oxygen and nutrients can effectively diffuse into living tissue. It is known that the threshold that oxygen can effectively diffuse through tissue is about 0.2 mm [31]. Considering that not only oxygen is needed to keep cells alive, but also nutrients, whose molecules are larger, the distance between capillaries and the necrotic core will be smaller than the mentioned 0.2 mm. Weinberg quotes the values 85 µm for human melanoma and 110 µm for rat prostate carcinoma [31,32]. We do not have similar data for cell lines under consideration; however, one of our estimated cases concerns B-16 cells, which is a murine melanoma tumour cell line used for research as a model for human skin cancers. The estimated kernel radius, although obtained for *in vitro* regimes, remains in good quantitative accordance with these data. Recall that $\sigma_k$ equal to 0.0342 corresponds to approximately 68 µm of a layer of viable cells. For L-5178Y and V-79 cell lines, obtained kernel size values correspond to 161 µm and 96 µm layer of viable cells, respectively. Following our theoretical consideration about the range of applicability of the proposed model, after its calibration against the experimental data, we postulate its suitability for describing proliferation in cell colonies whose growth is restricted to outer layers of viable cells. Let us mention that such a scenario is typical for most solid tumours, which, due to the lack of a regular blood vessels network typical for healthy tissues, develop its blood supply through the process of angiogenesis that results in a pathological capillary network producing numerous necrotic regions.

While analysing proliferation parameters for considered cell lines obtained with the MAP estimator, it becomes conspicuous that the value 1.7264 obtained for the L-5178Y cell line is noticeably larger than the values obtained for V-79 and B16 that are 0.3603 and 0.3616, respectively. The fact becomes more comprehensible if one considers also the dynamics of the entire colonies. Spheroids composed of L-5178Y cells grow much faster and reach a diameter of about 4 mm after only 30 days. The growth is

very fast but finishes shortly. What is more, the estimated value corresponds to the cell doubling time of about 10 h, which is exactly the value reported in databases [33]. In summary, it seems that in the initial growth of the L-5178Y spheroid, the presence of neighbourhood cells does not slow down the proliferation of L-5178Y cells. The proliferation parameters obtained for V-79 and B-16 cells are more similar to each other and equal to 0.3603 and 0.3616, respectively. More similar values of the results are not surprising as the spheroids composed of these cells also have more similar dynamics. Interestingly, the obtained values correspond to the division times for V-79 and B-16 cells which is approximately 46 h. This means a slowing down of the rate of divisions 2.8 times and 2.5 times, respectively. It is generally thought that cells in a colony are 2–3 times slower to divide. Our results for the V-79 and B-16 cells fit exactly into this framework.

Summing up our work, we state that comprehensive calibration of complex models describing the dynamics of the cancer disease *in vivo* seems for the moment to be out of reach, first of all, due to the lack of relevant data but also due to computational complexity of such tasks. Therefore, it seems appropriate to create at first partial, properly calibrated models that describe phenomena contributing to cancerogenesis and then combine them into more complex models to get more quantitative insight into the pathology of cancer development.

Data accessibility. All codes used to generate data within this manuscript have been uploaded as electronic supplementary material. Moreover, they are available at: https://github.com/Zuzanna-Szymanska/Non-local-proliferation-model and have been archived within the Zenodo repository: https://doi.org/10.5281/zenodo.5565314.

Authors' contributions. Z.S. proposed the proliferation model, selected data for Bayesian inference, reformulated the model in spherical coordinates and ran the simulations. B.M., Z.S. and P.G. developed the observation model. Z.S. and B.M. developed the code. J.S. and P.G. provided the analytical results on the convergence of measure solutions of the considered problem and proof of the stability of posterior distributions. Z.S. and J.S. wrote the manuscript. All authors approved the final version of the manuscript.

Competing interests. We declare we have no competing interests.

Funding. Z.S., B.M. and P.G. acknowledge the support from the National Science Centre, Poland, grant no. 2017/26/M/ST1/00783. J.S. was supported by the National Science Centre, Poland, grant no. 2019/35/N/ST1/03459. The calculations were made with the support of the Interdisciplinary Centre for Mathematical and Computational Modelling of the University of Warsaw under the computational grant no. G79-28.

Acknowledgements. All authors would like to express their gratitude to Michał Dzikowski and Bartosz Niezgódka from the Interdisciplinary Centre for Mathematical and Computational Modelling of the University of Warsaw for their valuable help in carrying out high-performance simulations.

# Appendix A. Lipschitz continuity of the inverse function

A simple lemma, which is in fact an inverse function theorem with a parameter. We use this lemma while proving that the inverse of the cumulative distribution function satisfies the continuity estimate. Recall that, if $f : \mathbb{R} \to (a, b)$ satisfies $f'(x) > 0$ then $f$ is globally invertible on $(a, b)$. Note that we are taking advantage of the fact that the problem under consideration is now one-dimensional since for multi-dimensional cases only local invertibility is true.

**Lemma A.1.** *Let $(S, d)$ be a metric space and $R \subset S$. Consider function $f(x, s) : (c, d) \times R \to (a, b)$ such that $f$ is differentiable with respect to $x$ and*

$$f_x(x, s) > \delta > 0. \tag{A 1}$$

*Let $f_s^{-1}$ be inverses of maps $x \mapsto f(x, s)$ with fixed $s$. Suppose that there are constants $C, D$ such that*

$$|f(x, s_1) - f(x, s_2)| \le C\,(d(s_1, s_2) + D).$$

*Then, for all $y \in (a, b)$,*

$$|f_{s_1}^{-1}(y) - f_{s_2}^{-1}(y)| \le \frac{C}{\delta}\,(d(s_1, s_2) + D).$$

*Proof.* Note that standard inverse function theorem implies that for all $s \in S$ we have $\|(f_s^{-1})'\|_\infty \leq \frac{1}{\delta}$. Hence, we can estimate

$$
\begin{aligned}
|f_{s_1}^{-1}(y) - f_{s_2}^{-1}(y)| &\leq |f_{s_1}^{-1}(f(f_{s_1}^{-1}(y), s_1)) - f_{s_1}^{-1}(f(f_{s_2}^{-1}(y), s_1))| \\
&\leq \frac{1}{\delta} |f(f_{s_1}^{-1}(y), s_1) - f(f_{s_2}^{-1}(y), s_1)| \\
&= \frac{1}{\delta} |f(f_{s_1}^{-1}(y), s_1) - f(f_{s_2}^{-1}(y), s_2) + f(f_{s_2}^{-1}(y), s_2) - f(f_{s_2}^{-1}(y), s_1)| \\
&= \frac{1}{\delta} |f(f_{s_2}^{-1}(y), s_2) - f(f_{s_2}^{-1}(y), s_1)| \leq \frac{C}{\delta} (d(s_1, s_2) + D).
\end{aligned}
$$

■

# Appendix B. Theory of measure solutions and convergence of particle method

Using the theory of measure approach for biological problems is becoming increasingly popular as it is very intuitive and convenient for both analytical and practical numerical simulation viewpoints. In general, measures assign real values to all measurable subsets of the considered space—we say that those real values are measures of sets. Naturally, non-negative measures are useful for describing various observable quantities, such as age, distribution or density. Importantly, the spaces of measures are vector spaces, which means in particular that the difference of two measures is again a measure from the same space. The numerical approximation we use to simulate the solutions to the model (2.3) is based on distributions not necessarily having densities with respect to the Lebesgue measure. Therefore, we reformulate our model for a generalized class of solutions in the space of non-negative Radon measures, cf. [34–40]. Within the new approach $\mu_t(A)$ is a measure such that, for every measurable set $A \in \mathbb{R}^+$, $\mu_t(A)$ is the mass of cells being at time $t$ at a distance from the centre of the coordinate system belonging to the set $A$. If $\mu_t$ has density $p(\cdot, t)$ then

$$
\mu_t(A) = \int_A p(r, t) \, \mathrm{d}r. \tag{B1}
$$

Nevertheless, our setting is more general and allows us to consider distributions that do not admit densities like combinations of Dirac masses. Using the EBT method, we approximate the solution to (2.3) assuming that each cohort represented by a Dirac mass corresponds to a mass of cells belonging to the set $A$, see (2.5).

We denote by $\mathcal{M}(\mathbb{R}^+)$ the space of all bounded and signed Radon measures on $\mathbb{R}^+$ whereas $\mathcal{M}^+(\mathbb{R}^+)$ stands for its subset, consisting of non-negative measures. Let us note that for $\mu \in \mathcal{M}(\mathbb{R}^+)$ we have unique Hahn–Jordan decomposition $\mu = \mu^+ - \mu^-$ where $\mu^+, \mu^- \in \mathcal{M}(\mathbb{R}^+)$. To work in the spaces of measures one needs the notion of a norm. The total variation norm is defined by

$$
\|\mu\|_{\mathrm{TV}} = \mu^+(\mathbb{R}^+) - \mu^-(\mathbb{R}^+), \tag{B2}
$$

which can be thought of as the total mass of $\mu$. Moreover, we denote with $\|\cdot\|_{BL^*}$ the flat norm

$$
\|\mu\|_{BL^*} := \sup\left\{ \int_{\mathbb{R}^+} \psi \, \mathrm{d}\mu : \psi \in BL(\mathbb{R}^+), \|\psi\|_{BL} \leq 1 \right\}, \tag{B3}
$$

where space of bounded Lipschitz functions $BL(\mathbb{R}^+)$ is given by

$$
BL(\mathbb{R}^+) = \{f : \mathbb{R}^+ \to \mathbb{R} \text{ is continuous and } \|f\|_\infty < \infty, \quad |f|_{Lip} < \infty\},
$$

and the relevant norms are defined as

$$
\|f\|_\infty = \sup_{x \in \mathbb{R}^+} |f(x)|, \quad |f|_{Lip} = \sup_{x \neq y} \frac{|f(x) - f(y)|}{d(x, y)}, \quad \|f\|_{BL} = \max(\|f\|_\infty, |f|_{Lip}).
$$

For stability properties of radial solutions, it is also important to introduce weighted flat norm given with

$$
\left\| \frac{\mu}{f(r)} \right\|_{BL^*} := \sup\left\{ \int_{\mathbb{R}^+} \frac{\psi(r)}{f(r)} \, \mathrm{d}\mu : \psi \in BL(\mathbb{R}^+), \|\psi\|_{BL} \leq 1 \right\}, \tag{B4}
$$

where $f(r)$ is a non-negative function. We refer to ([41], Ch 1) for all properties of metric space $(\mathcal{M}^+(\mathbb{R}^+), \|\cdot\|_{BL^*})$.

As we already mentioned, to prove the convergence of the employed numerical algorithm we embed the problem into the space of non-negative Radon measures. Since the concept is quite technical we refer the interested reader to our companion paper [6] to get the rigorous definition of a mild measure solution to (2.3); however, one may envisage a generalized solution in the sense of distributions. Below we present the theorem concerning the existence and uniqueness of measure solutions of (2.3).

**Theorem B.1.** *Let $\mu_0 \in \mathcal{M}^+(\mathbb{R}^+)$ be such that $\|\mu_0\|_{BL^*}$ and $\|\mu_0/r\|_{BL^*}$ are finite. Then, there exists a unique measure solution $\mu_t$ to (2.3) such that*

$$\sup_{t \in [0,T]} \|\mu_t\|_{BL^*} \leq \|\mu_0\|_{BL^*} e^{CT}$$

*and*

$$\sup_{t \in [0,T]} \left\| \frac{\mu_t}{r} \right\|_{BL^*} \leq \left\| \frac{\mu_0}{r} \right\|_{BL^*} e^{CT},$$

*where $C$ is a constant depending continuously on parameters $0 < \alpha$, $\sigma_k$, $\sigma_i$. Moreover, we have the following decay estimate: if $\mu_0$ satisfy $\int_{\mathbb{R}^+} e^x \, d\mu_0(x) < \infty$ then*

$$\int_{\mathbb{R}^+} e^x \, d\mu_t(x) \leq e^{Ct} \int_{\mathbb{R}^+} e^x \, d\mu_0(x). \tag{B5}$$

*In particular,*

$$\sup_{t \in [0,T]} \int_{(R_0,\infty)} d\mu_t \leq e^{CT} e^{-R_0}. \tag{B6}$$

**Remark B.2.** When $\mu_0$ is a radial measure, i.e. it is scaled with $r^2$ and compactly supported, all assumptions of theorem B.1 are satisfied.

**Remark B.3.** Tail estimate in (B6) allows assuming that the support of $\mu_t$ is finite.

**Remark B.4.** It is classical to apply flat norm in problems related to the convergence of particle method, cf. [36,40,42–44]. However, our kernel $L(R, r)$ defined by (2.4) is singular at $R = 0$ or $r = 0$ and therefore it does not satisfy the assumptions of the previous works. Despite this difficulty in our theoretical paper, we were able to prove convergence in the weighted flat norm [6].

We also recall the result on the convergence of the particle method and certain properties of particle approximation [6].

**Theorem B.5.** *Let $\mu_0$ be as in theorem B.1 and assume additionally that $\|\mu_0(r)/r^2\|_{BL^*} < \infty$. Consider its approximation $\mu_0^N = \sum_{i=1}^N m_i^N(0) \, \delta_{x_i}$ as defined in (2.7). Let $\mu_t^N = \sum_{i=1}^N m_i^N(t) \, \delta_{x_i}$, where $m_i^N(t)$ is the solution to the numerical scheme (2.6). Then,*

(A) *we have estimates*

$$\sup_{t \in [0,T]} \|\mu_t^N\|_{BL^*} \leq \|\mu_0\|_{BL^*} e^{CT}, \quad \sup_{t \in [0,T]} \left\| \frac{\mu_t^N}{r} \right\|_{BL^*} \leq \left\| \frac{\mu_0}{r} \right\|_{BL^*} e^{CT}$$

*where $C$ is a constant depending continuously on parameters $0 < \alpha$, $\sigma_k$, $\sigma_i$.*
(B) *$\mu_t^N$ satisfies the same decay bounds (B5)–(B6) as $\mu_t$.*

(C) *if $\mu_t$ is a measure solution to (2.3) with initial condition $\mu_0$ then for all $R_0 > 1$:*

$$\left\| \frac{\mu_t^N - \mu_t}{r} \right\|_{BL^*} \leq C \left[ \frac{R_0^2}{N} + e^{-R_0} \right]. \tag{B7}$$

We remark that in general one needs parameter $R_0$ to denote truncation of the support of solutions: in general, the solution is supported on the whole half-line even if the initial data is compactly supported.

# Appendix C. Simulated random walk Metropolis–Hastings algorithm

**Algorithm 1**. Random walk Metropolis–Hastings algorithm

---

**Result:** Sample $\{\theta_j\}_{j=0,..,M}$ from approximated posterior distribution

**Initialization:** $\theta_0 := [\log(\alpha^0), \log(\sigma_k^0), \log(\sigma_o^0), \log(\sigma_i)]$;

**for** $i = 0$ **to** $n$ **do**
  Compute $m_0(t_i)$;
**end**

**for** $j = 1$ **to** $M$ **do**
  $\tilde{\theta} \sim N(\theta_j, s \cdot \mathrm{Id})$;
  **for** $i = 0$ **to** $n$ **do**
    Compute $\tilde{m}(t_i)$
  **end**
  **if** $\mathcal{U}[0,1] \leq \min\left\{1, \frac{\pi(\tilde{\theta})\ell(D|\tilde{\theta})}{\pi(\theta_j)\ell(D|\theta_j)}\right\}$ **then**
    $\theta_j = \tilde{\theta}$
  **else**
    $\theta_j = \theta_{j-1}$
  **end**
**end**

---

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
