## [Peer Review File · Royal Society Open Science]

Review History

RSOS-211279.R0 (Original submission)

Review form: Reviewer 1

Is the manuscript scientifically sound in its present form?

Yes

Are the interpretations and conclusions justified by the results?

Yes

Is the language acceptable?

Yes

Do you have any ethical concerns with this paper?

No

Have you any concerns about statistical analyses in this paper?

No

Recommendation?

Accept with minor revision (please list in comments)

Comments to the Author(s)

For comments to the authors, please see attachment (Appendix A).

Review form: Reviewer 2

Is the manuscript scientifically sound in its present form?

Yes

Are the interpretations and conclusions justified by the results?

Yes

Is the language acceptable?

Yes

Do you have any ethical concerns with this paper?

No

Have you any concerns about statistical analyses in this paper?

No

Recommendation?

Accept with minor revision (please list in comments)

Comments to the Author(s)

The paper is well written and the research tackles a significant problem. The research article was able to acknowledge the current literature, identify the research gaps and contribute the . The article has necessary details in terms of statistical equations, methodological perspectives. I would suggest to add a supplementary material using the detailed steps of simulation, computation from experimental data and codes for reproducibility and replication reasons.

Decision letter (RSOS-211279.R0)

Dear Dr Szymańska

On behalf of the Editors, we are pleased to inform you that your Manuscript RSOS-211279 "BAYESIAN INFERENCE OF A NON-LOCAL PROLIFERATION MODEL" has been accepted for publication in Royal Society Open Science subject to minor revision in accordance with the referees' reports. Please find the referees' comments along with any feedback from the Editors below my signature.

Please submit your revised manuscript and required files (see below) no later than 7 days from today's (ie 22-Sep-2021) date. Note: the ScholarOne system will 'lock' if submission of the revision is attempted 7 or more days after the deadline. If you do not think you will be able to meet this deadline please contact the editorial office immediately.

on behalf of Dr Jose Carrillo (Associate Editor) and Mark Chaplain (Subject Editor)
openscience@royalsociety.org

Reviewer comments to Author:

Reviewer: 1

Comments to the Author(s)

For comments to the authors, please see attachment.

Reviewer: 2

Comments to the Author(s)

The paper is well written and the research tackles a significant problem. The research article was able to acknowledge the current literature, identify the research gaps and contribute the . The article has necessary details in terms of statistical equations, methodological perspectives. I would suggest to add a supplementary material using the detailed steps of simulation, computation from experimental data and codes for reproducibility and replication reasons.

===PREPARING YOUR MANUSCRIPT===

Your revised paper should include the changes requested by the referees and Editors of your manuscript. You should provide two versions of this manuscript and both versions must be provided in an editable format:
one version identifying all the changes that have been made (for instance, in coloured highlight, in bold text, or tracked changes);
a 'clean' version of the new manuscript that incorporates the changes made, but does not highlight them. This version will be used for typesetting.

Please ensure that you include an acknowledgements' section before your reference list/bibliography. This should acknowledge anyone who assisted with your work, but does not

qualify as an author per the guidelines at <https://royalsociety.org/journals/ethics-policies/openness/>.

===PREPARING YOUR REVISION IN SCHOLARONE===

-- Ensure that your data access statement meets the requirements at <https://royalsociety.org/journals/authors/author-guidelines/#data>. You should ensure that you cite the dataset in your reference list. If you have deposited data etc in the Dryad repository, please only include the 'For publication' link at this stage. You should remove the 'For review' link.

-- If you have uploaded ESM files, please ensure you follow the guidance at <https://royalsociety.org/journals/authors/author-guidelines/#supplementary-material> to include a suitable title and informative caption. An example of appropriate titling and captioning may be found at [https://figshare.com/articles/Table_S2_from_Is_there_a_trade-off_between_peak_performance_and_performance_breadth_across_temperatures_for_aerobic_sc ope_in_teleost_fishes_/3843624](https://figshare.com/articles/Table_S2_from_Is_there_a_trade-off_between_peak_performance_and_performance_breadth_across_temperatures_for_aerobic_scope_in_teleost_fishes_/3843624).

Author's Response to Decision Letter for (RSOS-211279.R0)

See Appendix B.

Decision letter (RSOS-211279.R1)

Dear Dr Szymańska,

I am pleased to inform you that your manuscript entitled "BAYESIAN INFERENCE OF A NON-LOCAL PROLIFERATION MODEL" is now accepted for publication in Royal Society Open Science.

on behalf of Dr Jose Carrillo (Associate Editor) and Mark Chaplain (Subject Editor)
openscience@royalsociety.org

Appendix A

Report for “Bayesian inference of a non-local proliferation model” submitted by Z. Szymanska, J. Skrzeczkowski, B. Miasojedow and P. Gwiazda

In this paper the authors propose a non-local model for proliferation instead of the well-studied logistic function combined with diffusion and taxis terms to capture the spatial expansion. The proposed proliferation model consists of a non-local logistic function where a convolution term is introduced to capture the emergence of new cells adjacent to proliferating cells.

To validate the model they use Bayesian inference for unknown parameters (an inverse problem approach) to show the accuracy of estimators on experimental data on multicellular spheroids.

I would like to recommend the acceptance of the paper after the following points have been considered.

Minor comments:

- Maybe I missed this in the paper, but is $\mu(A)$ defined somewhere? If not, I think one line such as $\mu(A) = \int_A u(x)dx$ where $u(x)$ is a density would be clarifying.
- In Theorem B.5 I didn't understand the third sentence. I guess μ_t^N is an approximation of the solution of (6) in the space of measures, where m_i solves (6). Could you clarify this, please?
- This is more of a personal question since I'm not an expert in Bayesian inference. In the first sentence in Section 5 it seems like you are performing Bayesian inference via MAP while these two approaches are very different. Could you clarify this? I recommend to re-phrase this sentence to avoid confusion.

Some typos:

- There is an extra *the* in p.4 line 17.
- Remove *that* in p.4 line 19.
- Remove *the* p.4 beginning of second paragraph.
- Final point in p.14 second equation.
- Comma at the end of the 6th equation in p.14.
- In theorem 4.9 “ $m_i(t)$ solveS THE system...”
- At beginning of Section 5 should be Appendix C instead of 3, p.17.

Appendix B

Replies to reviewers' comments on the manuscript "Bayesian inference of a non-local proliferation model"

by

Z. Szymańska, J. Skrzeczkowski, B. Miasojedow, and P. Gwiazda

We would like to thank the referees for their appreciation for our work, and their critical reading of this manuscript, and all their comments and questions. We have addressed all of the points raised by the referees. Our answers to them are provided below. We have marked in colour all changes done in the manuscript - apart from improvements suggested or provoked by the referees we also did a few very minor corrections like typos, punctuation marks, words adjustment etc.

Zuzanna Szymańska
Jakub Skrzeczkowski
Błażej Miasojedow
Piotr Gwiazda

Response to Referee 1:

- **Maybe I missed this in the paper, but is $\mu(A)$ defined somewhere? If not, I think one line such as $\mu(A) = \int_A u(x) dx$, where $u(x)$ is a density would be clarifying.**

Thank you very much for this remark, as the issue important and needs to be clarify. For measures $\mu(A)$ having density $u(x)$ indeed we have

$$\mu(A) = \int_A u(x) dx.$$

However, our setting is more general and allows us to consider distributions that do not admit densities like combinations of Dirac masses. We improved the relevant part in Appendix B, where we give the introduction on theory of measure solutions, see formula (27) and its description.

- **In Theorem B.5 I didn't understand the third sentence. I guess μ_t^N is an approximation of the solution of (6) in the space of measures, where m_i**

solves (6). Could you clarify this, please?

Yes, you are right. Somehow part of the sentence escaped from the right place. We have fixed it.

- **This is more of a personal question since I'm not an expert in Bayesian inference. In the first sentence in Section 5 it seems like you are performing Bayesian inference via MAP while these two approaches are very different. Could you clarify this? I recommend to re-phrase this sentence to avoid confusion.**

Typically the Bayesian inference is used with quadratic loss function which leads to posterior expectation as an estimator of unknown quantities. However, another approach is to use the MAP estimator which is especially useful when the posterior distribution is multimodal, which is in our case. We agree that mentioned sentence could be misleading and we have re-phrased it.

Some typos:

- **There is an extra the in p.4 line 17.**
- **Remove that in p.4 line 19.**
- **Remove the p.4 beginning of second paragraph.**
- **Final point in p.14 second equation.**
- **Comma at the end of the 6th equation in p.14.**
- **In theorem 4.9 " $m_i(t)$ solveS THE system..."**
- **At beginning of Section 5 should be Appendix C instead of 3, p.17.**

We have corrected all indicated typos.

Response to Referee 2:

The paper is well written and the research tackles a significant problem. The research article was able to acknowledge the current literature, identify the research gaps and contribute the . The article has necessary details in terms of statistical equations, methodological perspectives. I would suggest to add a supplementary material using the detailed steps of simulation, computation from experimental data and codes for reproducibility and replication reasons.

We would like to thank the reviewer for his appreciation for our work. We share with the reviewer the belief that all results published in the paper should be reproducible. Therefore, all the program codes used to prepare this article are available in our GitHub repository

<https://github.com/Zuzanna-Szymanska/Non-local-proliferation-model>

The codes are now better commented with precise information on the size of the measured data for all considered cell lines. The pseudocode of our Metropolis-Hastings algorithm is included in Appendix C.